# COVID-19 Vaccination in Multiple Sclerosis and Inflammatory Diseases: Effects from Disease-Modifying Therapy, Long-Term Seroprevalence and Breakthrough Infections

**DOI:** 10.3390/vaccines10050695

**Published:** 2022-04-28

**Authors:** Dejan Jakimovski, Karen Zakalik, Samreen Awan, Katelyn S. Kavak, Penny Pennington, David Hojnacki, Channa Kolb, Alexis A. Lizarraga, Svetlana P. Eckert, Rosila Sarrosa, Kamath Vineetha, Keith Edwards, Bianca Weinstock-Guttman

**Affiliations:** 1Buffalo Neuroimaging Analysis Center, Department of Neurology, Jacobs School of Medicine and Biomedical Sciences, University at Buffalo, State University of New York, Buffalo, NY 14203, USA; djakimovski@bnac.net; 2Jacobs Comprehensive MS Treatment and Research Center, Department of Neurology, Jacobs School of Medicine and Biomedical Sciences, University at Buffalo, Buffalo, NY 14202, USA; kzakalik@buffalo.edu (K.Z.); samreena@buffalo.edu (S.A.); kskavak@buffalo.edu (K.S.K.); ppennington99@gmail.com (P.P.); hojnacki@buffalo.edu (D.H.); cmkolb@buffalo.edu (C.K.); aalizarr@buffalo.edu (A.A.L.); svetlana@buffalo.edu (S.P.E.); 3MS Center of Northeastern NY—Empire Neurology, P.C., Latham, NY 12110, USA; rsarrosa@tristateneuro.com (R.S.); vkamath@tristateneuro.com (K.V.); kedwards@tristateneuro.com (K.E.)

**Keywords:** COVID-19, SARS-CoV-2 vaccination, BNT162b2, mRNA-1273, Ad26.COV2.S, breakthrough infection, DMT

## Abstract

Background: To determine the effect of disease-modifying therapies (DMT) on humoral postvaccine seroconversion, long-term humoral response, and breakthrough COVID-19 infections in persons with multiple sclerosis (PwMS) and other neuroinflammatory disorders. Methods: A total of 757 PwMS and other neuroinflammatory disorders were recruited in two MS centers and vaccinated with one of the FDA-approved vaccines (BNT162b2, mRNA-1273, Ad26.COV2.S). The primary outcomes are the rate of humoral postvaccine seroconversion and anti-severe acute respiratory syndrome coronavirus 2 (anti-SARS-CoV-2) immunoglobulin G (IgG) differences between patients on different DMTs. Secondary measures include breakthrough infections and humoral response after six months. Other outcomes include differences in vaccine response between SARS-CoV-2 vaccines and the effects of age and comorbidities on the vaccine response. Results: A total of 465 (68.4%) PwMS and 55 (74.3%) patients with neuroinflammatory diseases were seropositive at 4–12 weeks after vaccination. A significant difference in seroconversion based on the DMT used at the time of vaccination (*p* < 0.001) was observed, with the lowest rates seen in patients treated with anti-CD20 antibodies (23.2%) and sphingosine-1-phosphate modulators (S1P) (30.8%). In seropositive patients, there was a significant decrease in anti-SARS IgG from mean 20.0 to 4.7 at six months (*p* = 0.004). Thirty-nine patients had breakthrough infection, but only two seronegative patients required hospitalization. mRNA vaccines resulted in significantly greater seroconversion compared to Ad26.COV2.S (*p* < 0.001). Older age and presence of cardiovascular comorbidities were associated with lower anti-SARS IgG (*p* = 0.021 and *p* = 0.003, respectively) Conclusions: PwMS and neuroinflammatory disorders treated with anti-CD20 and S1P medications have lower humoral response after anti-SARS-CoV-2 vaccination, even after booster dose. Waning of the humoral response puts vaccinated PwMS at a greater risk of COVID-19 breakthrough.

## 1. Introduction

Since the start of the severe acute respiratory syndrome coronavirus 2 (SARS-CoV-2) pandemic, more than 250 million recorded cases and more than 5 million COVID-19-related deaths have been recorded worldwide. Older patients, people with pre-existing cardiovascular comorbidities and patients on immunosuppressive therapy have been at particular risk of severe COVID-19 outcome [1]. A global effort to develop and distribute SARS-CoV-2 vaccines has resulted in the manufacturing of at least 25 different vaccines (of which eight have been approved by the world health organization). 

Persons with multiple sclerosis (PwMS) are at increased risk of severe COVID-19 disease, due to their immunocompromised status (medication-induced), as well as to age, disability, and high prevalence of cardiovascular comorbidities [2]. A recent meta-analysis showed that PwMS are at significantly higher risk of severe COVID-19 outcomes and COVID-19-related mortality when compared to the general population [3]. Unvaccinated PwMS treated with anti-CD20 monoclonal antibodies (rituximab and ocrelizumab) are significantly more likely to be admitted to intensive care units and have greater mortality [4]. Moreover, immunosuppressive MS disease modifying therapies (DMTs) could decrease the benefit derived from vaccines. The lack of sufficient humoral response after vaccination has previously been documented [5,6].

Based on this background, we aimed to investigate the effect of DMTs on SARS-CoV-2 vaccine seroconversion in a group of PwMS or other neuroinflammatory diseases (such as neuromyelitis optica spectrum disorder (NMOSD), autoimmune encephalitis, isolated transverse myelitis, systemic inflammatory disease with CNS involvement and unspecified demyelinating disorders). More importantly, we investigated factors influencing the extent of humoral response from the three available vaccines, including the length from a previous drug administration, presence of comorbidities, and patient age. Lastly, the durability of the humoral response and the occurrence of breakthrough infections were measured at six-months post-vaccination.

## 2. Materials and Methods

### 2.1. Study Population

PwMS from two New York State Multiple Sclerosis Consortium (NYSMSC) sites (Buffalo, NY, USA and Albany, NY, USA) were enrolled into this study, as well as patients with other neuroinflammatory disorders enrolled from Buffalo, NY. Study inclusion criteria included: (1) MS diagnosis as per 2017-revision of the McDonald criteria [7] or presence of other neuroinflammatory disease; and (2) willingness to undergo COVID-19 antibody testing 4–12 weeks after vaccination. The DMTs used by the PwMS were classified into categories based on the drug’s mechanism of action. In particular, both subcutaneous and intramuscular interferon (IFN)-β-1a, IFN-β-1b and peg-IFN-β-1a were classified as IFN-β medications. Rituximab, ocrelizumab and ofatumumab were all grouped under the category of anti-CD20 monoclonal antibodies (mAb). Fingolimod, ponesimod, siponimod and ozanimod were grouped under the sphingosine-1-phopshate (S1P) category. Both dimethyl fumarate and diroximel fumarate were grouped under fumarates and all glatiramer acetate (GA) products (Copaxone^®^, generic glatiramer acetate, 20 mg or 40 mg) were combined. Lastly, off-label MS medications included cyclophosphamide, intravenous immunoglobulin, azathioprine, methotrexate and mycophenolate mofetil. Data regarding patients’ comorbidities (diabetes, dyslipidemia, hypertension, and obesity) were also collected.

As per approval from the U.S. Food and Drug Administration (FDA), the study population received one of three vaccines: BNT162b2 (Pfizer, New York City, NY, USA), mRNA-1273 (Moderna, Cambridge, MA, USA) and Ad26.COV2.S (Johnson & Johnson, New Brunswick, NJ, USA). Patients provided further information about the occurrence of any immediate or late-onset vaccine reactions. The seroconversion to the vaccination was determined using commercially available and FDA-emergency authorized assays such as the Luminescence-based VITROS anti-SARS-CoV-2 IgG test (VITROS ECi/ECiQ/3600 Immunodiagnostic Systems, Rochester, NY, USA) and anti-SARS-CoV-2 Semi-Quantitative Total Antibody 164090 (LabCorp, Burlington, NC, USA). Based on each test cut-off value (<0.4, <1.0 or <1.4), patients were classified as seropositive or seronegative. During the COVID-19 pandemic, the antibody testing cut-off values (and the assay used) were changed by the manufacturer itself, where each change did not affect the equivalency of the findings. Concretely, previously seroconverted patient with cut-off index >1.0 was translated as positive with cut-off of >1.4. For all anti-SARS IgG index analyses, the seronegative patients were categorically labeled and did not enter the numerical analyses. Patients with anti-SARS IgG index above 20 were classified as high IgG category and patients with IgG index between 1 and 20 were classified as low IgG category.

When available, data regarding white blood cell count (WBC), absolute lymphocyte count (ALC), serum protein electrophoresis (SPEP)-derived gamma globulin levels, and CD19+ cell count were collected before the vaccination, between 4–12-weeks after vaccination, and six months after vaccination. SARS-CoV-2 infections supported by positive polymerase chain reaction (PCR) test were recorded if they preceded the vaccination, or as breakthrough infections when they occurred after vaccination. 

### 2.2. Statistical Analyses

All statistical analyses were performed using SPSS version 26.0 (IBM, Armonk, NY, USA) and data was visualized in GraphPad Prism version 8.0 (San Diego, CA, USA). Data distribution was assessed using Kolmogorov-Smirnov test of normality and through visual inspection of the data histograms and Q-Q plots. Group comparisons were performed with Student’s *t*-test, Mann-Whitney U-test, and pair-wise *t*-test when applicable, where categorical data was compared using χ^2^ test. One-way analysis of variance (ANOVA) and Kruskal Wallis H-test were used as well. Age and disease duration-adjusted logistic regression models were utilized as well. No imputing for missing data was performed and cases with missing values were excluded for each respective analysis. *p*-values lower than 0.05 were considered statistically significant. 

## 3. Results

### 3.1. Demographic and Clinical Characteristics of the Study Populations

The demographic and clinical characteristics are shown in Table 1. In total, 757 patients were recruited (683 PwMS and 74 with other neuroinflammatory disorders). Based on sites, 614 were recruited in Buffalo, NY and 143 in Albany, NY. There were no age, sex or race differences between the PwMS and other neuroinflammatory disorders. The 74 patients with neuroinflammatory diseases were: (1) 22 NMOSD (ICD code G36.0), (2) 20 unspecified demyelinating disease (ICD code G37.9) (3) 3 transverse myelitis (ICD code G37.3), (4) 1 optic neuritis (ICD code H46.9), and (5) other neuroinflammatory disorders such as autoimmune encephalitis, autoimmune vasculitis, peripheral autoimmune polyneuropathy, acute demyelinating encephalomyelitis and chronic inflammatory demyelinating polyradiculoneuropathy. PwMS had neurological symptoms at a younger age (36 vs. 41.8 years old, *p* < 0.001). Similarly, no differences were reported in the type of COVID-19 vaccine used (*p* = 0.41) or in the percentage of patients reporting symptoms after the vaccination (*p* = 0.315). As expected, there were significant differences in the type of medications between the two groups. 

Among PwMS, 361 (52.9%) were vaccinated with BNT152b2, 271 (39.7%) were vaccinated with mRNA-1273 and 49 (7.2%) with Ad26.COV2.S. A small percentage of PwMS (5.3%) reported mild, short-lasting neurological symptoms after the vaccination. The most common DMT used by PwMS were anti-CD20 mAb group (25.9%), followed by IFN-β (14.6%), GA (9.2%), fumarates (9.1%), natalizumab (8.8%), teriflunomide (5.6%), S1P modulators (3.8%), off-label medications (2.6%), cladribine (1.9%) and alemtuzumab (0.7%). The remaining 121 (17.7%) were not treated with any DMT.

Subcutaneous and intramuscular interferon (IFN)-β-1a, IFN-β-1b and peg-IFN-β-1a were classified as IFN-β medications. All glatiramer acetate variates (Copaxone^®^, generic glatiramer acetate, 20 mg or 40 mg) were grouped together. Rituximab, ocrelizumab and ofatumumab were all grouped under the category of Anti-CD20 mAb. Fingolimod, ponesimod, siponimod and ozanimod were grouped under the S1P category. Both dimethyl fumarate and diroximel fumarate were grouped under Fumarates. Off-label MS medications included cyclophosphamide, intravenous immunoglobulin, azathioprine, methotrexate and mycophenolic acid.

### 3.2. DMT Effect on COVID-19 Vaccine Seroconversion

Before vaccination, 30 patients were diagnosed with COVID-19. After vaccination, 71.9% of them had seroconversion and presence of anti-SARS IgG antibodies. The overall seroconversion to COVID-19 vaccination in PwMS and other neuroinflammatory population is shown in Table 2. In total, 465 (68.4%) out of 683 PwMS seroconverted after COVID-19 vaccination. Similarly, 55 (74.3%) out of 74 patients with other neuroinflammatory diseases seroconverted. Out of the 237 patients with no seroconversion, 32 were based on the later <1.4 cut-off value (13.5%), only 5 with the original <0.4 cut-off (2.1%) and all remaining 200 cases were scored using the <1.0 cut-off value. There were no disproportions on cut-off value use based on the DMT of the patient (*p* > 0.05). There were no significant differences in the rate of seroconversion between male and female (*p* = 0.17) where 114 out of 177 (64.8%) males were seropositive and 406 out of 580 (70.2%) females were seropositive. There was a significant difference in seroconversion based on the DMT used at the time of vaccination (*p* < 0.001), where successful seroconversion was noted in 85.0% of patients on IFN-β, 88.9% in patients on GA, 87.1% in patients on fumarates, 73.7% in patients on teriflunomide, 98.3% in patients on natalizumab, 88.9% in patients on off-label DMT and 61.5% in patients on cladribine. Lower seroconversion rates were seen in patients treated with S1P products (30.8%) and anti-CD20 mAbs (23.2%) (all seroconversion rates are shown in Figure 1). All five patients treated with alemtuzumab had successful COVID-19 vaccine seroconversion. Similar findings were seen in patients with other neuroinflammatory diseases, with 20% seroconversion in patients treated with anti-CD20 mAbs. 

The differences in seroconversion between the different DMT groups remained significant when both male and female patients were investigated separately (significant DMT effect of *p* < 0.001). There was a significant age difference between patients treated with different DMTs (one-way ANOVA *p* < 0.001). In particular, patients treated with S1Ps were the youngest, with an average age of 47.8 years old, followed by fumarates (49.3 years old), natalizumab (50.0 years old) and anti-CD20 mAbs (52.1 years old). In contrast, patients treated with IFN-β, GA and no DMT were among the oldest participants (60.2, 59.7 and 59.3 years old, respectively). In an adjusted logistic regression analysis (for age and age at diagnosis), the DMT effect on seroconversion remained statistically significant (Negelkerke R^2^ = 0.305 for the entire model, with DMT *p* < 0.001).

Seroconversion analysis based on the vaccine used is shown in Table 3. Patients vaccinated with Ad26.COV2.S had significantly lower percentage of seroconversion when compared to mRNA-based vaccines (*p* = 0.005 for IFN-β, p=0.019 for GA, *p* < 0.001 for patients on fumarates and *p* = 0.01 for patients on anti-CD20 mAbs). Albeit not statistically significant, patients vaccinated with mRNA-1273 had relatively greater percentage of seroconversion when compared to BNT162b2 (IFN-β 95.3% vs. 82.2%, teriflunomide 82.4% vs. 72.2%, S1Ps 37.5% vs. 27.8%, cladribine 75% vs. 55.6% and anti-CD20 mAbs 36.5% vs. 16.2%). Differences between vaccines was further confirmed through anti-SARS IgG index where Ad26.COV2.S had significantly lower levels when compared to mRNA-based vaccines and mRNA-1273 had numerically greater IgG index when compared to BNT162b2 (*p* < 0.023, Appendix A). 

Seroconversion outcomes based on the time since the latest anti-CD20 mAb infusion and effects from the DMT history were further analyzed. PwMS that seroconverted had received their vaccine at a significantly later time from the last anti-CD20 mAb infusion compared to seronegative PwMS (6.5 months vs. 4.6 months, *p* = 0.001) Moreover, there was a significant correlation between the anti-SARS IgG index after seroconversion and the time from the latest anti-CD20 mAb infusion and the second dose (r_s_ = 0.239, *p* = 0.014). There was no relationship between the time of latest cladribine or alemtuzumab exposure and seroconversion (average time since alemtuzumab exposure to vaccination was 3.8 years, and the average time since cladribine exposure to vaccination was 147.9 days). When compared to all seropositive patients, the seronegative patients more commonly had a prior DMT history which includes use of immunosuppressive S1P, anti-CD20 mAbs and off-label DMT medications (2.7% vs. 7.8% for prior S1P use, 5.9% vs. 13.3% for natalizumab, 2.5% vs. 7.2% for off-label DMTs and 5.9% vs. 9.0% for anti-CD20 mAb, χ^2^ test *p* < 0.001). Albeit not significant, a similar trend was seen in an analysis which excluded patients that were currently treated with S1Ps or anti-CD20 mAbs, where seronegative patients had numerically greater history of immunosuppression (*p* = 0.197). 

### 3.3. Effects of Age, Comorbidities and Prior SARS-CoV-2 Infection on Vaccine Seroconversion

There were no significant age differences between patients with positive seroconversion compared to patients without seroconversion after the vaccination course (56.0 vs. 54.4 years, *p* = 0.125). Similar findings were seen for each of the vaccines investigated separately (*p* = 0.702 for BNT162b2, *p* = 0.173 for mRNA-1273, and *p* = 0.363 for Ad26.COV2.S). However, patients not on DMT and no seroconversion after vaccination were significantly older when compared to seroconverters (65.4 vs. 58.5 years *p* = 0.008). Similar findings were seen in patients treated with S1Ps (50.6 vs. 41.4 years, *p* = 0.051). In seroconverted non-DMT patients, higher age was significantly associated with lower absolute anti-SARS IgG index (r_s_ = −0.198, *p* = 0.021). 

There was no difference in the percentage of seroconversion between patients without or with comorbidities (*p* = 0.562). However, non-DMT patients with comorbidities had significantly lower seroconversion than non-DMT patients without any comorbidity (80.3% vs. 96.3%, *p* = 0.003). Similar findings were seen in comorbid vs. non-comorbid patients treated with S1Ps (0% vs. 32.0%. *p* = 0.032) and anti-CD20 mAbs (20.0% vs. 38.9%, *p* = 0.032). 

Before vaccine availability and/or administration, 33 patients were diagnosed with COVID-19. All 33 were later vaccinated (21 with BNT162b2, 9 with mRNA-1273, 2 with Ad26.COV2.S, 1 unknown). When tested after vaccination, 71.9% were seropositive and 90.5% had high IgG index (>20 IgG Index). The remaining 9 seronegative patients were mainly treated with anti-CD20 mAbs (7 patients, 77.9%), one patient treated with cladribine and one not on any DMT (11.1% each). Proportionately, these seronegative patients after prior COVID-19 and vaccination were as follows: 7 out of 11 (63.6%) of patients on anti-CD20 mAbs, 1 out of 6 (16.7%) not on any DMT and 1 out 1 (100%) cladribine-treated patient.

### 3.4. White Cell Count, Absolute Lymphocyte Count, CD19+ Cell Levels and Vaccine Seroconversion

Differences in WBC, ALC, SPEP-based gamma globulin levels and CD19+ count between patients with and without seroconversion are shown in Table 4. These measures were performed before receiving the vaccine and 4–12 weeks after full vaccination course. Seronegative patients had significantly lower pre-vaccination ALC (1415.3 vs. 1785.9 cells, *p* < 0.001) and lower CD19+ (54.6 vs. 225.4 cells, *p* < 0.001) when compared to seropositive. The same findings were seen in measures 4–12-weeks after vaccination. The significant findings regarding the effect of low ALC and CD19+ levels were also present for both BNT162b2 and mRNA-1273. Due to insufficient sample size, this analysis was not performed for Ad26.COV2.S. 

Student’s *t*-test was used. *p*-values lower than 0.05 were considered statistically significant and shown in bold. The ALC findings from the total study population also investigated in DMT categories outside of the S1Ps and anti-CD20 mAb patients. Seronegative patients treated with IFN-β had significantly lower ALC when compared seropositive patients (1308.8 vs. 1632.7 cells, *p* = 0.041). No significant findings were seen in the remaining groups, mainly due to low number of seronegative patients. The CD19+ finding was specific to S1Ps and anti-CD20 mAb medications. 

### 3.5. Six-Month Seroprevalence and IgG Index Stability in Multiple Sclerosis

Overall, 125 PwMS were re-tested six months after their vaccination. The majority of cases maintained their early classification, i.e., 83 previously positive cases remained positive at six months and 21 previously negative cases remained negative. On the other hand, nine early positive cases had no more detectable antibodies at six months. 

Lastly, 12 initially negative cases tested positive after a six-month re-evaluation (a significant difference in result distribution between both timepoints, *p* < 0.001). Eight of these patients were vaccinated with BNT162b2 and four with mRNA-1273. Five of the 12 patients received their third booster dose before their six-month follow-up and one patient was diagnosed with a breakthrough infection before their six-month follow-up. The remaining cases could be explained by an asymptomatic infection not captured by PCR testing.

There was a significant decrease in absolute anti-SARS IgG index from median 20.0 at 4–12 weeks to median 4.7 at six months (Wilcoxon Signed ranks test Z = −4.143, *p* < 0.001). Changes in IgG index based on the vaccine used are shown in Appendix A. The most significant decrease in IgG index occurred in patients vaccinated with BNT162b2, i.e., from median 11.7 to median 16.7 (Wilcoxon Signed ranks test Z = −2.9, *p* = 0.004), followed by patients vaccinated with mRNA-1272, with median values increasing from 17.1 to 19.6 (Wilcoxon Signed ranks test Z = −2.45, *p* = 0.014). No significant decline in index values were observed in patients vaccinated with Ad26.COV2.S: median 2.4 to 0.7 (Wilcoxon Signed ranks test Z = −1.402, *p* = 0.161). 

### 3.6. Booster Dose and Seroconversion

At the time of data analysis, 239 PwMS had received a third booster dose and 121 of them underwent new antibody testing after the booster dose. One hundred and three (85.1%) of PwMS with booster dose had seroconversion and 18 PwMS (14.9%) did not. The lack of seroconversion occurred only in PwMS treated with anti-CD20 mAbs (13 PwMS, 72.2%), and S1P (4 PwMS, 22.2%). Only one remaining patient not treated on any DMT was seronegative after the booster dose. (χ^2^
*p* < 0.001). Proportionately, 13 out of 21 (61.9%) of patients on anti-CD20 mAbs were seronegative after a booster dose, 4 out of 8 (50%) S1Ps treated patients were seronegative after booster dose and 1 out of 27 (3.7%) of non-DMT-treated patients were seronegative after booster dose. When compared to the seroconversion after the second dose, out of 37 previously negative PwMS, 20 (54.4%) did develop antibody response after the booster dose. The DMT of these PwMS ranged from anti-CD20 mAbs (51.4%), S1Ps (13.5%), teriflunomide (10.8%), IFN-β and off-label DMTs (8.1% each), two PwMS on no DMT (5.4%) and one PwMS on fumarate (2.7%). Lastly, only one previously seropositive PwMS did not have detectable antibodies after the booster dose.

### 3.7. Breakthrough Infections after Vaccination

After vaccination, 39 patients were diagnosed with COVID-19 and all patients recovered from the infection. The breakthrough infections occurred, on average, 6.6 months after the second vaccine dose and 7 out of 39 breakthrough infections after receiving the boost. Overall, there was a numerically greater rate of breakthrough infection in seronegative when compared to seropositive patients (8.6% vs. 4.9%, *p* = 0.088). The majority of PwMS with breakthrough infection were treated with either anti-CD20 mediation (41%), followed by natalizumab (12.8%), glatiramer acetate (12.8%), fumarates (10.3%) and S1Ps (7.7%). Anti-spike antibody analysis after the vaccination and preceding the infection showed 22 (56.4%) were seropositive and 14 (35.9%) were seronegative and no information was available for 3 patients. Proportionately, the highest rate of breakthrough infections was within patients on anti-CD20 mAbs (16 out of 117; 13.7%), natalizumab (5 out of 38, 13.7%) and S1Ps (3 out of 25, 12%). In contrast, the smallest ratio was seen in patients not treated on any DMT (2 out of 148, 1.4%), IFN-β (2 out of 100, 2%) and off-label DMTs (1 out of 43, 2.3%). Within these breakthrough COVID-19 cases there was no differences in which mRNA vaccine was used (53.8% vaccinated with BNT162b2 and 41% with mRNA-1273 and 2% with Ad26.COV.2.S). Additional two SARS-CoV-2 infections occurred in-between BNT162b2 doses (both having high anti-SARS IgG in follow-up testing). 

Eight patients after PCR confirmation sought additional medical care and received anti-COVID-19 treatment (antibodies such as bamlanivimab/etesevimab, casirivimab/imdevimab and sotrovimab) with good recovery. One seronegative patient on anti-CD20 therapy developed severe COVID-19 pneumonia that required hospitalization and oxygen support. On the other hand, a seropositive patient (with high IgG index) treated on glatiramer acetate was also hospitalized and treated for COVID-19 pneumonia. 

## 4. Discussion

The findings from this prospective observational SARS-CoV-2 vaccine response study are multifold. First, PwMS treated with anti-CD20 mAbs or S1Ps have significantly lower seroconversion rate when compared to patients treated with other medications. This effect was present regardless of which SARS-CoV-2 vaccine was used. Secondly, lower ALC and CD19+ cell count at the time of vaccination was significantly associated with lower seroconversion. This finding may be specific to the S1Ps and anti-CD20 mAb-treated patients. Thirdly, patients vaccinated with mRNA vaccines had significantly better seroconversion compared to Ad26.COV.2.S (mRNA-1273>BNT162b2). Without a booster dose, the humoral response, as measured by anti-spike antibodies, wanes significantly after six months, and breakthrough infections in vaccinated patients (being both seropositive or seronegative) can occur. While boosters provide increased seroconversion in previously seronegative patients, PwMS treated with anti-CD20 mAbs and S1Ps remain negative, even after their third dose.

The pattern of low humoral response to SARS-CoV-2 vaccination in patients treated with anti-CD20 mAbs and S1Ps has been previously reported in the literature. Some reports have calculated that in comparison to non-treated PwMS, patients on ocrelizumab have as much as 201-fold decrease in post-vaccination antibodies [8]. The 26-fold decrease in patients treated with fingolimod and 20-fold decrease in patients treated with rituximab was also recorded. Similar findings were seen in smaller British and French cohorts [9,10]. We also corroborate that mRNA-1273 does elicit higher immunogenicity when compared to BNT162b2 (3.25 times higher levels as reported in the CovaXiMS study) [8]. Lastly, we also corroborated the findings that close timing between anti-CD20 mAbs infusion and vaccination significantly decrease the humoral response [11,12]. Therefore, clinically stable PwMS may consider delaying the anti-CD20 infusion in favor of greater likelihood of COVID-19 vaccine seroconversion.

The waning humoral response (decline in anti-SARS IgG antibodies) was also documented in a large BNT162b2-vaccinated population, where the participants had a sharp decline after three months post-vaccination and remained low at the six-month mark [13,14]. We further corroborate their results of age-dependency and immunosuppression effect [13]. Similar DMT effects on six-month vaccine response were also seen in an Israel-based study, where patients treated with fingolimod and ocrelizumab showed decreased seropositivity [15]. There are contradictory findings in the literature regarding the effect of comorbidities [16,17]. While higher BMI has been associated with a greater level of neutralizing antibodies, we showed that presence of a cardiovascular comorbidity may be associated with lower seroconversion. One reason for the differences may stem from how the data are captured (absolute anti-SARS IgG findings versus categorial seroconversion rate) and whether seronegative patients contribute to the IgG analysis. The humoral response measurements could only depict part of the total immune response. SARS-CoV-2-specific T-cell response could provide sufficient protection toward severe COVID-19 outcomes, and may explain the heterogeneity of our results and the data in the literature [18,19]. Currently, there are no data available regarding the sustained T-cell response after SARS-CoV-2 vaccination and the relative amount of protection related to the T-cell response.

In our study, 5.3% of the patients reported mild and short lasting neurological adverse events after vaccination. These findings may be attributed to overlapping symptoms from the disease, although the overall reactogenicity to SARS-CoV-2 vaccination in PwMS is not significantly different than in the general population. In a recent study of 719 patients, 64% reported some adverse reaction and 17% reported a severe reaction 24 hours after receiving a vaccine dose [20]. Interestingly, there were significant discrepancies in the reactogenicity to the vaccine based on the DMT used, with patients treated with natalizumab and S1Ps reporting lower adverse reactions [20]. Late adverse events to the SARS-CoV-2 vaccination (particularly with the three US-approved vaccines) are extremely rare, with only few case series and reports being published [21,22].

The seroconversion rate and IgG index are of particular importance in PwMS due to the risk for breakthrough SARS-CoV-2 infections. Most breakthrough infections occurred later in the follow-up period when the IgG levels were already declining. Although most of our PwMS had relatively mild breakthrough COVID-19, there were two patients with severe outcome which required hospitalization. The severity of the COVID-19 infection in one of our hospitalized patients can be attributed to the failure to seroconvert, mainly as a result of the use of anti-CD20 mAb, older age (65 years old) and the presence of cardiovascular comorbidity. The second hospitalized patient was also older and several cardiovascular comorbidities which may contributed to severe COVID-19 outcome despite the significant vaccine seroconversion. A recent report which followed 1497 vaccinated medical professionals showed that 39 people had a breakthrough infection [23]. Low neutralizing antibodies measured during peri-infectious phase was outlined as one potential feature for the people that had breakthrough infection. As outlined in our results, the breakthrough infections in these medical professionals were mainly mild, but persistent symptoms (>6 weeks) seen in 19% [23]. More importantly, people with breakthrough infections have low infectivity and do not contribute to the spread of the virus [23]. An epidemiological survey and a detailed description of a COVID-19 breakthrough infection in one of our patients can be found elsewhere [24].

While we describe that a third of previously seronegative patients became seropositive after their booster dose, a group of anti-CD20 mAb and S1Ps treated PwMS still remained seronegative. With the relatively high prevalence of breakthrough infections, and failure of seroconversion after the third vaccine dose, PwMS taking these classes of DMTs are advised to receive additional fourth dose. Further analysis and data regarding whether PwMS treated with anti-CD20 mAbs and S1Ps should continue with additional doses or discontinue therapy during vaccination is warranted. Lastly, future studies should determine whether the level of ALC and CD19+ cells are drug-specific or these findings are applicable to the general population. 

One particular limitation in this study was a lack of direct healthy control comparison. However, the rate of seroconversion in healthy individuals has been previously reported, and was deemed significantly higher when compared to our results. For example, 95.9% of individuals younger than 60 years old had positive serology after 21 days from only one BNT152b2 dose [25]. Similar findings were seen with mRNA-1273 where all healthy individuals (with no prior SARS-CoV-2 exposure) developed successful seroconversion [26]. Another limitation was the use of commercial quantitative immunoenzymatically assays which have changed their cut-off values by the later stages of our recruitment. That said, only a small and equally distributed percentage of our patients (4.9%) were tested using modified cut-off values, thus not affecting the DMT findings. 

## 5. Conclusions

In conclusion, PwMS treated with S1P modulators and anti-CD20 mAbs have poor humoral response after SARS-CoV-2 vaccination. The seroconversion and extent of humoral response can be influenced by age, presence of comorbidities, time since previous immunosuppressive infusion and lymphocyte levels. Most PwMS have relative waning of the humoral response after 6 months from vaccination. Lack of seroconversion and strong humoral response may increase the risk for breakthrough SARS-CoV-2 infections. While booster doses increase the overall seroprevalence in the PwMS population, PwMS treated with anti-CD20 mAbs and S1Ps still exhibit lower humoral response.

## Figures and Tables

**Figure 1 vaccines-10-00695-f001:**
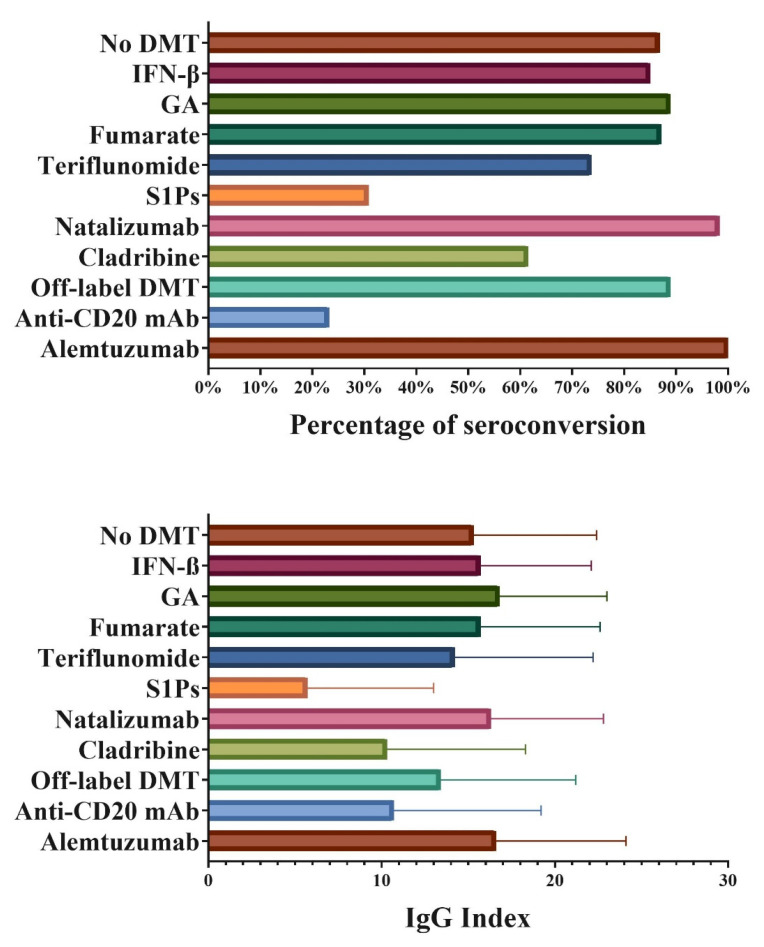
COVID-19 vaccine seroconversion and IgG index in seropositive patients based on disease modifying therapy. **Legend:** DMT—disease modifying therapy, IFN—interferon, S1Ps—sphingosine-1-phosphate, IgG—immunoglobulin G. Rituximab, ocrelizumab and ofatumumab were all grouped under the category of Anti-CD20 mAb. Fingolimod, ponesimod, siponimod and ozanimod were grouped under the S1P category. Both dimethyl fumarate and diroximel fumarate were grouped under Fumarates. Off-label MS medications included cyclophosphamide, intravenous immunoglobulin, azathioprine, methotrexate and mycophenolic acid.

**Table 1 vaccines-10-00695-t001:** Demographic and clinical characteristics of the study populations.

Demographic and Clinical Characteristics	Total MS Population (n = 683)	Other Diagnoses (n = 74)	*p*-Value
Buffalo, NY	540 (79.1)	74 (100.0)	-
Albany, NY *	143 (20.9)	-	
Female, n (%)	529 (77.5)	51 (68.9)	0.099
Caucasian, n (%)	490 (90.7)	63 (85.1)	**0.043**
Age of symptom onset, mean (SD)	36.0 (11.5)	41.8 (17.1)	**<0.001**
Vaccine type, n (%)			0.41
BNT162b2	361 (52.9)	35 (47.3)	
mRNA-1273	271 (39.7)	36 (48.6)	
Ad26.COV2.S	49 (7.2)	3 (4.1)	
Symptoms after vaccination, n (%)			0.315
Yes	28 (5.3)	7 (9.6)	
No	406 (76.2)	52 (71.2)	
Unknown	99 (18.6)	14 (19.2)	
DMT at vaccination, n (%)			**<0.001**
No DMT	121 (17.7)	34 (45.9)	
IFN-β	100 (14.6)	0 (0.0)	
Glatiramer acetate	63 (9.2)	0 (0.0)	
Fumarate	62 (9.1)	0 (0.0)	
Teriflunomide	38 (5.6)	0 (0.0)	
S1Ps	26 (3.8)	0 (0.0)	
Natalizumab	60 (8.8)	0 (0.0)	
Cladribine	13 (1.9)	0 (0.0)	
Off-label DMT	18 (2.6)	25 (33.8)	
Anti-CD20 mAb	177 (25.9)	15 (20.3)	
Alemtuzumab	5 (0.7)	0 (0.0)	

**Legend:** MS—multiple sclerosis, DMT—disease modifying therapy, IFN—interferon, S1Ps—sphingosine-1-phosphate, SD—standard deviation, mAb—monoclonal antibody. Statistically significant *p*-values lower than 0.05 are shown in bold. * Data regarding race, disease duration and symptoms after vaccination were not available for the patients recruited in Albany, NY.

**Table 2 vaccines-10-00695-t002:** Seroconversion after COVID-19 vaccination based on disease modifying therapy.

PwMS (n = 683)	Seroconversion	High IgG Index Category	Low IgG Index Category	IgG Index	Other Population (n = 74)	Seroconversion	High IgG Index Category	Low IgG Index Category	IgG Index
No DMT	105 (86.8)	68 (69.4)	30 (30.6)	15.3 (7.1)	No DMT	33 (97.1)	26 (78.8)	7 (21.2)	17.2 (5.8)
IFN-β	85 (85.0)	55 (67.1)	27 (32.9)	15.7 (6.4)	Off-label DMT	19 (76.0)	9 (47.4)	10 (52.6)	10.9 (8.5)
GA	56 (88.9)	39 (72.2)	15 (27.8)	16.8 (6.2)	Anti-CD20 mAb	3 (20.0)	1 (33.3)	2 (66.7)	9.5 (9.6)
Fumarate	54 (87.1)	32 (69.6)	14 (30.4)	15.7 (6.9)	*p*-value	**<0.001**	**0.035**		0.53
Teriflunomide	28 (73.7)	17 (68.0)	8 (32.0)	14.2 (8.0)					
S1Ps	8 (30.8)	2 (28.6)	5 (71.4)	5.7 (7.3)					
Natalizumab	59 (98.3)	41 (71.9)	16 (28.1)	16.3 (6.5)					
Cladribine	8 (61.5)	2 (33.3)	4 (66.7)	10.3 (8.0)					
Off-label DMT	16 (88.9)	8 (53.3)	7 (46.7)	13.4 (7.8)					
Anti-CD20 mAb	41 (23.2)	20 (51.3)	19 (48.7)	10.7 (8.5)					
Alemtuzumab	5 (100)	4 (80.0)	1 (20.0)	16.6 (7.5)					
*p*-value	**<0.001**	0.124		0.31					

**Legend:** PwMS—persons with multiple sclerosis, DMT—disease modifying therapy, IFN—interferon, S1Ps—sphingosine-1-phosphate, SD—standard deviation, mAb—monoclonal antibody. Statistically significant *p*-values are shown in bold. Rituximab, ocrelizumab and ofatumumab were all grouped under the category of Anti-CD20 mAb. Fingolimod, ponesimod, siponimod and ozanimod were grouped under the S1P category. Both dimethyl fumarate and diroximel fumarate were grouped under Fumarates. Off-label MS medications included cyclophosphamide, intravenous immunoglobulin, azathioprine, methotrexate and mycophenolic acid. Patients with anti-SARS IgG index above 20 were classified as high IgG category and patients with IgG index between 1/1.4 and 20 were classified as low IgG category. A similar DMT trend was seen on anti-SARS IgG index in seropositive positive patients. Patients treated with S1Ps, cladribine and anti-CD20 mAbs were more commonly categorized within the low IgG index group and had numerically lower absolute IgG index (5.7 for S1Ps, 10.3 for cladribine and 10.7 for anti-CD20 mAbs) (shown in Figure 1). *p*-values lower than 0.05 were considered statistically significant and shown in bold.

**Table 3 vaccines-10-00695-t003:** Seroconversion and IgG index in PwMS based on COVID-19 vaccine.

Vaccine	BNT162b2		mRNA-1273		Ad26.COV2.S		
Seroconversion	Seropositive	Seronegative	Seropositive	Seronegative	Seropositive	Seronegative	*p*-Value
No DMT	51 (86.4)	8 (13.6)	47 (90.4)	5 (9.6)	7 (70.0)	3 (30.0)	0.218
IFN-β	37 (82.2)	8 (17.8)	41 (95.3)	2 (4.7)	7 (58.3)	5 (41.7)	**0.005**
GA	27 (93.1)	2 (6.9)	27 (93.1)	2 (6.9)	2 (50.0)	2 (50.0)	**0.019**
Fumarate	31 (93.9)	2 (6.1)	23 (92.0)	2 (8.0)	0 (0.0)	3 (100.0)	**<0.001**
Teriflunomide	13 (72.2)	5 (27.8)	14 (82.4)	3 (17.6)	1 (33.3)	2 (66.7)	0.202
S1Ps	5 (27.8)	13 (72.2)	3 (37.5)	5 (62.5)	-	-	0.62
Natalizumab	33 (100)	0 (0.0)	21 (95.5)	1 (4.5)	5 (100.0)	0 (0.0)	0.415
Cladribine	5 (55.6)	4 (44.4)	3 (75.0)	1 (25.0)	-	-	0.506
Off-label DMT	9 (81.8)	2 (18.2)	7 (100.0)	0 (0.0)	-	-	0.231
Anti-CD20 mAb	16 (16.2)	83 (83.8)	23 (36.5)	40 (63.5)	1 (8.3)	11 (91.7)	**0.01**
Alemtuzumab	4 (100)	0 (0.0)	1 (100.0)	0 (0.0)	-	-	-

**Legend:** MS—multiple sclerosis, DMT—disease modifying therapy, IFN—interferon, S1Ps—sphingosine-1-phosphate, SD—standard deviation, mAb—monoclonal antibody. Statistically significant p-values are shown in bold. Rituximab, ocrelizumab and ofatumumab were all grouped under the category of Anti-CD20 mAb. Fingolimod, ponesimod, siponimod and ozanimod were grouped under the S1P category. Both dimethyl fumarate and diroximel fumarate were grouped under Fumarates. Off-label MS medications included cyclophosphamide, intravenous immunoglobulin, azathioprine, methotrexate and mycophenolic acid. *p*-values lower than 0.05 were considered statistically significant and shown in bold.

**Table 4 vaccines-10-00695-t004:** Effects of white blood cell count and absolute lymphocyte count on seroconversion to COVID-19 vaccination.

Pre-Vaccination Status	Available Samples	Seroconversion	No Seroconversion	*p*-Value
WBC	433 vs. 163	6.5 (3.3)	6.0 (2.2)	0.123
ALC	392 vs. 148	1785.9 (823.1)	1415.3 (723.7)	**<0.001**
SPEP—gamma globulin	75 vs. 43	1.1 (0.4)	0.9 (0.4)	0.179
CD19 + cells	140 vs. 97	225.4 (222.9)	54.6 (166.9)	**<0.001**

**Legend:** WBC—white blood count, ALC—absolute lymphocyte count, SPEP—serum protein electrophoresis, CD19—cluster of differentiation 19 (marker for B-cells). WBC is shown as 10e9/µL, ALC in absolute cell number in µL, SPEP as amount of gamma globulin in grams per deciliter (g/dL). *p*-values lower than 0.05 were considered statistically significant and shown in bold.

## Data Availability

The data that support the findings of this study are available from the corresponding author upon reasonable request.

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
