# Peer review of "COVID-19 Vaccination in Multiple Sclerosis and Inflammatory Diseases: Effects from Disease-Modifying Therapy, Long-Term Seroprevalence and Breakthrough Infections"

_vaccines, 2022, doi:10.3390/vaccines10050695_

Round 1
Reviewer 1 Report
The manuscript by Jakimowski et al. describes the effects of anti SARS-CoV-2 vaccinations in a vast population of Multiple Sclerosis patients (PwMS).The topic is far from original, despite some of the treatments considered in the study had been scarcely investigated before (in terms of their effects on vaccination) as the number of patients analyzed in this cohort is significantly higher compared to other studies. However, the study shows 2 important scientific flaws:
- There is no real control population, which should have been a population of healthy individuals comparable (in term of numbers) to that of PwMS
- The authors rely on results obtained by 2 different qualitative immunoenzymatic tests, without further detail on how their use was distributed in the population, and how their sensitivity and specificity compared to each other. In addition, they inappropriately use the “index” (apparently even indexes from two different tests as if they were the same!) as a reliable quantification unit, which of course is prone to artifacts, as in uncalibrated qualitative tests such as those mentioned by the authors, the index does not closely follow the real binding power of the serum, especially for high values such as those obtained after vaccination.
These issues significantly reduce the value of the study. In addition, there is an evident lack of originality, as all the conclusions the authors come to had already been exhaustively described in a score of previous studies.
Author Response
The manuscript by Jakimowski et al. describes the effects of anti SARS-CoV-2 vaccinations in a vast population of Multiple Sclerosis patients (PwMS).The topic is far from original, despite some of the treatments considered in the study had been scarcely investigated before (in terms of their effects on vaccination) as the number of patients analyzed in this cohort is significantly higher compared to other studies. However, the study shows 2 important scientific flaws:
- There is no real control population, which should have been a population of healthy individuals comparable (in term of numbers) to that of PwMS
- The authors rely on results obtained by 2 different qualitative immunoenzymatic tests, without further detail on how their use was distributed in the population, and how their sensitivity and specificity compared to each other. In addition, they inappropriately use the “index” (apparently even indexes from two different tests as if they were the same!) as a reliable quantification unit, which of course is prone to artifacts, as in uncalibrated qualitative tests such as those mentioned by the authors, the index does not closely follow the real binding power of the serum, especially for high values such as those obtained after vaccination.
These issues significantly reduce the value of the study. In addition, there is an evident lack of originality, as all the conclusions the authors come to had already been exhaustively described in a score of previous studies.
- We thank the Reviewer for the comments. We have expanded our Discussion to address both limitations that were outlined. In particular, seroconversion in healthy individuals has been extensively published as part of the original vaccine trials and showed almost 100% successful seroconversion. These findings were significantly higher than the seroconversion seen in our PwMS population.
Jackson LA, Anderson EJ, Rouphael NG, et al. An mRNA Vaccine against SARS-CoV-2 - Preliminary Report. N Engl J Med 2020;383:1920-1931.
Shachor-Meyouhas Y, Hussein K, Szwarcwort-Cohen M, et al. Single BNT162b2 vaccine dose produces seroconversion in under 60 s cohort. Vaccine 2021;39:6902-6906.
- We have further explained the distribution of each test in the study population. Only small proportion of the total population (37 out of 757 participants; 4.9%) were tested with an assay that was not using the cut-off index of <1.0. In particular, only 5 subjects were classified as negative based on the <0.4 cut-off and 32 subjects based on the later <1.4 cut-off value. There were no DMT biases in terms of which cut-off was used for any particular medication/response outcome. Moreover, the manufactures of the immunoassays have declared equivalency of the findings when their cut-offs were changed. (They informed us that the classification of negative vs. positive remains true for all previous and new outcomes, despite the change in cut-off value). This was further explained as a limitation in our manuscript.
Reviewer 2 Report
In the present article, Jakimovski et al. provide interesting findings that are relevant to the field of multiple sclerosis and give important information about the interaction between SARS CoV2 vaccination and neuroinflammatory diseases treatments.
The manuscript reads well and the figures are clear, I only have a few minor comments:
- Acronyms should be spelled out and avoided in the abstract (MS, SARS)
- What diseases were included in "other diagnoses"? do these inflammatory diseases belong to the same group of diseases? More information should be provided, at least a list of disease and how many patients.
- The infection is by SARS-CoV2 not COVID19.
- While the influence of age and weight were looked what about sex? Did it affect the response?
- How CD19 level was measured? is it a cell count of CD19+ cells or a protein level?
- Line 293: After vaccination is repeated in the same sentence.
- Would anti SARS IgG more accurate than SARS IgG
Author Response
In the present article, Jakimovski et al. provide interesting findings that are relevant to the field of multiple sclerosis and give important information about the interaction between SARS CoV2 vaccination and neuroinflammatory diseases treatments.
The manuscript reads well and the figures are clear, I only have a few minor comments:
- We thank the Reviewer for the positive comments. We provide point-by-point responses to his minor comments shown hereafter:
1) Acronyms should be spelled out and avoided in the abstract (MS, SARS)
- Both acronyms have been spelled out as suggested.
2) What diseases were included in "other diagnoses"? do these inflammatory diseases belong to the same group of diseases? More information should be provided, at least a list of disease and how many patients.
- We have provided a more comprehensive breakdown of the diseases included in the other non-MS neuroinflammatory diseases.
3) The infection is by SARS-CoV2 not COVID19.
- This has been corrected throughout the manuscript.
4) While the influence of age and weight were looked what about sex? Did it affect the response?
- Unfortunately, we did not systematically collect the BMI of the patients in this particular study. We did include a comment on the BMI effect in the Discussion of the manuscript demonstrating that patients with greater BMI had a greater level of neutralizing antibodies. We further expanded on the sex analysis in the Results. There were no differences in the rate of seroconversion between the genders (p=0.17). The effect of DMT on seroconversion was similar in both females and males separately. (both p<0.001)
5) How CD19 level was measured? is it a cell count of CD19+ cells or a protein level?
- These were cells measured using a classical flow cytometer as part of routine cell count measurements. We specified CD19+ cell count throughout the manuscript.
6) Line 293: After vaccination is repeated in the same sentence.
- This has been corrected.
7) Would anti SARS IgG more accurate than SARS IgG
- We agree with the Reviewer. This has been corrected throughout the manuscript.
Reviewer 3 Report
This article aims at assessing serological response to COVID19 vaccination in patients with MS as well as other various inflammatory diseases of the nervous system.
While this topic has already been explored in some previous publications, confirming those data on a large-size cohort is interesting. Interestingly, authors also provide data about breakthrough infections occurring after vaccination, which is highly relevant even if it occurred in a small number of patients.
The results are consistent with what has already been reported, showing a decreased serological response in patients treated with antiCD20 therapies as well as S1P modulators. Interestingly, immunogenicity seemed better with mRNA-1273 vaccine in comparison with BNT162b2, which had already been suggested in COVAXIMS study and could have important consequences in our practice.
Here are my comments
We can see in the table 1 that patient groups were not balanced regarding DMT distribution, particularly for MS patients , with a high representation of patients treated with anti CD20. Did the author compare the demographics of patients according to their DMT? Regarding age and physical disability which can be confounding factors)
Moreover, several times, authors compare the absolute number of a given event in different group of patients according to their DMT. For example regarding breakthrough infections: “The majority of PwMS with breakthrough infection were treated with either anti-CD20 mediation (41%), followed by natalizumab (12.8%)…”. As antiCD20 was the most employed DMT in the cohort, it is not surprising to see that this kind of event occurred more frequently in this group. I think it could have been more relevant to provide the event rate in each DMT group (XX% of patients under antiCD20 therapies experienced breakthrough infection, XX% in the natalizumab group, etc…). This remarks applies to several parts of the results section.
Regarding the breakthrough infections, if I understood well, more than 50% of those events occurred in patients who had a good serological response and only 36% in patients who were seronegative. One again, it would be much more relevant to know the global rate of breakthrough infections in the seropositive versus seronegative group to have a better idea of how important seroconversion is to predict the risk of reinfection
Regarding immune reconstitution therapies, even if it doesn’t represent a large number of patients, could the authors precise the delay between last treatment exposure and vaccination?
About the discussion:
Authors suggest that “lower ALC and CD19+ count at the time of vaccination was significantly associated with 316 lower seroconversion”. Does this finding only reflect the fact that seroconversion rate was lower for patients treated with S1P modulators (inducing low ALC in a large majority of patients) and antiCD20 therapies (inducing prolonged depletion of CD19+ cells) or is it a real independent predictive factor? Could the author explore this question? If no, it should at least be discussed
Author Response
This article aims at assessing serological response to COVID19 vaccination in patients with MS as well as other various inflammatory diseases of the nervous system.
While this topic has already been explored in some previous publications, confirming those data on a large-size cohort is interesting. Interestingly, authors also provide data about breakthrough infections occurring after vaccination, which is highly relevant even if it occurred in a small number of patients.
The results are consistent with what has already been reported, showing a decreased serological response in patients treated with antiCD20 therapies as well as S1P modulators. Interestingly, immunogenicity seemed better with mRNA-1273 vaccine in comparison with BNT162b2, which had already been suggested in COVAXIMS study and could have important consequences in our practice.
- We thank the Reviewer for the positive comments. We provide a point-by-point responses to the comments.
Here are my comments
We can see in the table 1 that patient groups were not balanced regarding DMT distribution, particularly for MS patients , with a high representation of patients treated with anti CD20. Did the author compare the demographics of patients according to their DMT? Regarding age and physical disability which can be confounding factors)
- We thank the Reviewer for the excellent suggestion. We included the age of each DMT group and performed the age and age of diagnosis-adjusted logistic regression model regarding the relationship between seroconversion and DMT In general, the patients that were on more immunosuppressive therapies (S1Ps and anti-CD20) were significantly younger than the population treated on other DMTs. These findings suggest that the effect of DMTs on seroconversion is significant despite the younger age of these patients.
Moreover, several times, authors compare the absolute number of a given event in different group of patients according to their DMT. For example regarding breakthrough infections: “The majority of PwMS with breakthrough infection were treated with either anti-CD20 mediation (41%), followed by natalizumab (12.8%)…”. As antiCD20 was the most employed DMT in the cohort, it is not surprising to see that this kind of event occurred more frequently in this group. I think it could have been more relevant to provide the event rate in each DMT group (XX% of patients under antiCD20 therapies experienced breakthrough infection, XX% in the natalizumab group, etc…). This remarks applies to several parts of the results section.
- We agree with the Reviewer. This has been corrected in the manuscript and we also addressed the ratios in several aspects of the Results section. The same conclusions can be derived from these findings where proportionately, the highest rate of breakthrough infections was among patients on anti-CD20 mAbs (16 out of 117; 13.7%), natalizumab (5 out of 38, 13.7%) and S1Ps (3 out of 25, 12%). Contrarily, the smallest ratio was seen in patients not treated on any DMT (2 out of 148, 1.4%), IFN-β (2 out of 100, 2%), and off-label DMTs (1 out of 43, 2.3%).
- In a similar fashion, we created proportional findings for the booster dose where 13 out of 21 (61.9%) of patients on anti-CD20 mAbs were seronegative after a booster dose, 4 out of 8 (50%) S1Ps treated patients were seronegative after booster dose and 1 out of 27 (3.7%) of non-DMT-treated patients were seronegative after the booster dose.
- This was also implemented in patients with a diagnosis of COVID-19 before vaccination.
Regarding the breakthrough infections, if I understood well, more than 50% of those events occurred in patients who had a good serological response and only 36% in patients who were seronegative. One again, it would be much more relevant to know the global rate of breakthrough infections in the seropositive versus seronegative group to have a better idea of how important seroconversion is to predict the risk of reinfection
- We agree with the Reviewer and provided these statistics in the manuscript. Overall, there was a numerically greater rate of breakthrough infection in seronegative when compared to seropositive patients (8.6% vs. 4.9%, p=0.088).
Regarding immune reconstitution therapies, even if it doesn’t represent a large number of patients, could the authors precise the delay between last treatment exposure and vaccination?
- In addition to the anti-CD20 mAb data, we also included the time from the latest alemtuzumab and cladribine exposure and vaccination. There was no relationship between the time of latest cladribine or alemtuzumab exposure and seroconversion. (average time since alemtuzumab exposure to vaccination was 3.8 years and the average time since cladribine exposure to vaccination was 147.9 days).
About the discussion:
Authors suggest that “lower ALC and CD19+ count at the time of vaccination was significantly associated with 316 lower seroconversion”. Does this finding only reflect the fact that seroconversion rate was lower for patients treated with S1P modulators (inducing low ALC in a large majority of patients) and antiCD20 therapies (inducing prolonged depletion of CD19+ cells) or is it a real independent predictive factor? Could the author explore this question? If no, it should at least be discussed
- We have performed the ALC and CD19+ analysis in each study group (S1Ps and anti-CD20 mAbs) and in other DMT groups (particularly in patients not on any DMTs). We included a short explanation in the Discussion as well.
Round 2
Reviewer 3 Report
All my comments have been addressed thank you very much